# Validation of an ICH Q2 Compliant Flow Cytometry-Based Assay for the Assessment of the Inhibitory Potential of Mesenchymal Stromal Cells on T Cell Proliferation

**DOI:** 10.3390/cells12060850

**Published:** 2023-03-09

**Authors:** Natascha Piede, Melanie Bremm, Anne Farken, Lisa-Marie Pfeffermann, Claudia Cappel, Halvard Bonig, Theres Fingerhut, Laura Puth, Kathrin Vogelsang, Andreas Peinelt, Rolf Marschalek, Matthias Müller, Peter Bader, Zyrafete Kuçi, Selim Kuçi, Sabine Huenecke

**Affiliations:** 1Division for Stem Cell Transplantation, Immunology and Intensive Care Medicine, Department for Children and Adolescents, University Hospital, Goethe University Frankfurt, Theodor-Stern-Kai 7, 60590 Frankfurt am Main, Germany; 2German Red Cross Blood Service BaWüHe, Institute Frankfurt, Sandhofstrasse 1, 60528 Frankfurt am Main, Germany; 3Institute for Transfusion Medicine and Immunohematology, Goethe University Frankfurt, Sandhofstrasse 1, 60528 Frankfurt am Main, Germany; 4Institute of Pharmaceutical Biology, Goethe University Frankfurt, Max-von-Laue-Strasse 9, 60438 Frankfurt am Main, Germany; 5Department Pharmaceutical Research & Development, Medac Gesellschaft für Klinische Spezialpräparate mbH, Theaterstrasse 6, 22880 Wedel, Germany

**Keywords:** mesenchymal stem/stromal cell, lymphocyte proliferation, potency assay, quality control, mixed lymphocyte reaction, graft-versus-host disease, flow cytometry, MSC manufacturing

## Abstract

Mesenchymal stromal cells (MSCs) have the potential to suppress pathological activation of immune cells and have therefore been considered for the treatment of Graft-versus-Host-Disease. The clinical application of MSCs requires a process validation to ensure consistent quality. A flow cytometry-based mixed lymphocyte reaction (MLR) was developed to analyse the inhibitory effect of MSCs on T cell proliferation. Monoclonal antibodies were used to stimulate T cell expansion and determine the effect of MSCs after four days of co-culture based on proliferation tracking with the violet proliferation dye VPD450. Following the guidelines of the International Council for Harmonisation (ICH) Q2 (R1), the performance of *n* = 30 peripheral blood mononuclear cell (PBMC) donor pairs was assessed. The specific inhibition of T cells by viable MSCs was determined and precision values of <10% variation for repeatability and <15% for intermediate precision were found. Compared to a non-compendial reference method, a linear correlation of r = 0.9021 was shown. Serial dilution experiments demonstrated a linear range for PBMC:MSC ratios from 1:1 to 1:0.01. The assay was unaffected by PBMC inter-donor variability. In conclusion, the presented MLR can be used as part of quality control tests for the validation of MSCs as a clinical product.

## 1. Introduction

Mesenchymal stromal cells (MSCs) are a heterogeneous adherent population of non-haematopoietic multipotent progenitor cells, which are characterized by fibroblast-like morphology, adherent growth and trilineage differentiation potential towards osteoblasts, chondrocytes and adipocytes [1,2,3]. Several clinical studies showed the potential of MSCs as an immunotherapy to treat steroid-refractory graft-versus-Host-Disease (GvHD) following allogeneic hematopoietic stem cell transplantation (HSCT) [4,5,6,7,8,9]. Third-party MSCs suppress the proliferation of activated lymphocytes in a non-HLA (human leucocyte antigen)-restricted manner [6,10]. Here, MSCs pursue different pathways to alleviate GvHD such as direct cell–cell contact, suppression of cytokine production, secretion of specific soluble factors and up-regulation of inhibitory surface receptors [3,11]. It has been shown that there are considerable differences in MSC efficacy and safety in terms of donor heterogeneity, tissue origin or manufacturing process [12,13]. With the goal of providing quickly accessible batches (“off-the-shelf”) of pharmaceutical-grade MSC products with consistent allosuppressive potential, a new protocol for MSC generation was developed, which is based on pooling bone marrow mononuclear cells from eight different donors, and an MSC bank was generated [6]. To enable the screening and ensure high immunosuppressive potency of MSCs intended for clinical application, a robust International Council for Harmonisation (ICH) Q2 compliant in vitro pharmaceutical potency assay is required [14,15].

In GvHD pathology, T cells contained in the graft recognize HLA-disparate molecules of the host as non-self, which results in T cell activation and proliferation. The alloreactive cells attack the patient’s body and cause an inflammatory cascade leading to widespread tissue damage predominantly in epithelial tissues [16]. Mixed lymphocyte reaction (MLR) assays simulate T cell activation similar to what occurs in a patient’s body during active GvHD by in vitro mixing of allogeneic lymphocyte populations and measuring the subsequent alloresponse in the form of T cell proliferation [17]. The activation requires the engagement of the CD3 receptor on T cells by antigen-presenting cells and a second co-stimulatory signal through the CD28 receptor. This process can be mimicked or enhanced using anti-CD3 and anti-CD28 antibodies in vitro [18]. MLR assays have been used for several years to study the immunosuppressive potential of drug candidates [19], checkpoint inhibitors [20] and third-party donor cells [10]. Therefore, different detection methods can be used to quantify lymphocyte proliferation including the measurement of DNA-synthesis [21,22], metabolic activity [23,24] or fluorescence-activated cell staining (FACS)-based proliferation tracking [25,26,27,28,29].

In our MLR assay, purified anti-CD3 and anti-CD28 antibodies are used to specifically stimulate T cell proliferation. The extent of cell expansion is detected with a violet proliferation dye (VPD450), which is partly passed to daughter generations after each cell division. After a period of four days, proliferation can be read out with flow cytometry. Furthermore, 7AAD is used to distinguish living from dead cells, and specific markers for T cell identification have been applied.

To employ this MLR assay for the pharmacological qualification of the allosuppressive potential of clinical-grade MSCs, the MLR itself must be validated for compliance with the regulations of the local authorities. In order to provide an assay that allows the critical assessment of MSC potency, a flow cytometry-based MLR was developed according to the guidelines of the European Pharmacopoiea, which are described in the guidelines of the ICH Q2 (R1) [14,15,30]. We propose the application of the MLR as part of the quality control release tests for the qualification of good manufacturing practice (GMP)-grade MSC products to sensitively identify subpotent batches and prevent their release.

## 2. Materials and Methods

### 2.1. Isolation of MSCs and PBMCs

MSCs were generated from pooled bone marrow mononuclear cells of eight healthy adult donors as described before [6]. The clinical-grade MSC drug products were provided by Medac GmbH in frozen cryobags. Peripheral blood mononuclear cells (PBMCs) were derived from buffy coats from anonymized healthy adult volunteers collected by the German Red Cross Blood Donor Service Baden-Württemberg–Hessen, Frankfurt, with written informed consent and permission of the local ethics committee (vote #329/10). PBMCs were isolated using density gradient centrifugation (Cat# P04-60500, PAN Biotech, Aidenbach, Germany), washed twice with phosphate-buffered saline (PBS, Cat# 14190094, Gibco, Darmstadt, Germany) and pipetted through a 70 µm cell strainer (Cat# 352350, BD Falcon, Heidelberg, Germany). PBMC purity and cell counts were determined using the DxH 500 haematology analyser (Beckman Coulter, Krefeld, Germany).

### 2.2. PBMC Labelling for Cell Proliferation Tracking

Isolated PBMCs from two donors were mixed in equal parts and stained for subsequent cell proliferation tracking using VPD450 (Cat# 562158, BD Horizon, Heidelberg, Germany). To find out the optimal dye concentration, we performed a titration of VPD450, which was reconstituted according to the manufacturer’s instructions and added to the PBMC donor pairs. Cells were stained for 10 min at 37 °C with constant shaking (>70 rpm). After two subsequent washing steps with PBS and one final washing step with Roswell Park Memorial Institute Medium 1640 (RPMI-1640 + GlutaMAX^TM^ Supplement, Cat# 61870036, Gibco, Darmstadt, Germany) and 10% fetal bovine serum (FBS, Cat# F7524, Sigma-Aldrich, Taufkirchen, Germany), PBMCs were resuspended in a co-culture medium consisting of RPMI + GlutaMAX^TM^ and 10% FBS.

### 2.3. Stimulated Co-Culture of PBMCs and MSCs

Frozen MSCs were thawed for 2–3 min in the plasmatherm Cell & Gene (Barkey, Leopoldshoehe, Germany) at 37 °C. Cells were then mixed with pre-cooled MSC medium consisting of DMEM (Dulbecco’s Modified Eagle Medium, Cat# 21885025, Gibco, Darmstadt, Germany) supplemented with 5 IU Heparin/mL (PZN-03029820, Ratiopharm, Ulm, Germny) and 10% human thrombocyte lysate (provided by the German Red Cross Blood Donor Service Baden-Württemberg–Hessen, Frankfurt). MSC number and viability were determined using trypan blue exclusion staining (Cat# T8154, Sigma-Aldrich, Taufkirchen, Germany). MSCs were then mitotically inactivated using gamma irradiation (30 Gy) and seeded at 0.3 × 10^6^ cells/cm^2^. After 2 h of letting the cells adhere, a final concentration of 0.5 × 10^6^ PBMCs/mL was added (1:1 ratio). Additionally, 1:0.2, 1:0.1 and 1:0.05 ratios of PBMC:MSC were prepared. The T cell stimulants Ultra-LEAF™ purified anti-human CD3 antibody (Cat# 300332, BioLegend, Amsterdam, The Netherlands) and Ultra-LEAF™ purified anti-human CD28 antibody (Cat# 302934, BioLegend, Amsterdam, The Netherlands) were added at a final concentration of 0.4 µg/mL each. Each co-culture condition was prepared in triplicates. For the FACS-based MLR, cells were incubated for four days in 48-well plates (Cat# 353078, Corning, Kennebunk, ME, USA). For the Bromodeoxyuridine (BrdU) MLR, which has been applied as a non-compendial reference assay, opaque 96-well plates (Cat# 3916, Corning, Kennebunk, ME, USA) were used, and co-culture was performed for 7 days. Where indicated, proliferation was analysed after 1, 2, 3, 5 or 6 days instead.

### 2.4. Flow Cytometry Analysis

A six-colour flow cytometry panel was designed to specifically analyse the proliferation of T cells. Therefore, cells were pipetted through a 40 µm cell strainer (Cat# 43-10040-70, pluriSelect, Leipzig, Germany) and stained with CD45 FITC (Cat# A07782, Beckman Coulter, Krefeld, Germany), 7-AAD (Cat# A07704, Beckman Coulter, Krefeld, Germany), CD4 PC7 (Cat# 737660, Beckman Coulter, Krefeld, Germany), CD8 PC7 (Cat# 737661, Beckman Coulter, Krefeld, Germany) and CD5 APC (Cat# 345783, BD Biosciences, Heidelberg, Germany). For PBMCs freshly isolated on day 0, cells were incubated with 1 mL of 1× NH_4_Cl-based erythrocyte lysing solution (contained in Cat# IM3630d, Beckman Coulter) for 10 min at room temperature. Analysis was performed using the FACSLyric Flow Cytometer and the FACSuite software 1.4 (BD Biosciences, Heidelberg, Germany).

In order to determine the extent of T cell proliferation, a gating strategy was established for CD45^+^/7AAD^−^/CD5^+^/CD4^+^ or CD8^+^ singlet cells. Those cells are further referred to as T cells. Then, the proportion of T cells that had divided at least once (fraction diluted, Dil as described by Roederer et al. [31]) was determined and the number of daughter generations arising from the initial population was counted. Mean values for each triplicate were calculated to determine the inhibition by MSCs using the following formula:Inhibition = 100% − (Dil_[co-culture]_/Dil_[stimulated PBMC]_ × 100)(1)

### 2.5. BrdU Assay

In order to assess the extent of PBMC proliferation, the BrdU Cell Proliferation ELISA Kit (Cat# 11669915001, Roche, Penzberg, Germany) was used according to the manufacturer’s instructions. Briefly, 10 µM BrdU labelling solution was added on day six. After 24 h of incubation, the plates were dried for 1–2 h at 60 °C. The fixation and denaturation reagent was added for 30 min at room temperature and was afterwards replaced with an anti-BrdU detection reagent. After 90 min incubation at room temperature, the cells were washed three times and the BrdU substrate solution was added. Chemiluminescence detection was performed for 1 s per well and immediately after 300 s of shaking using the VICTOR^3^ multilabel reader and the Wallac 1420 Manager Software version 3.0 (Perkin Elmer, Rodgau, Germany). During the whole procedure, the cells were protected from light. To determine the percentage inhibition of proliferation, the relative luminescence units (RLUs) measured with the device were used. For each condition, the mean value was calculated and the inhibition by MSCs was determined using the following formula:Inhibition = 100% − (RLU_[co-culture]_/RLU_[stimulated PBMC]_ × 100)(2)

### 2.6. Statistical Evaluation and Validation of the Method

For the validation of an analytical method, the ICH Q2 (R1) guidelines specify eight parameters that must be considered [14]: (1) specificity, (2) accuracy, (3) precision, (4) detection limit, (5) quantitation limit, (6) linearity, (7) range and (8) robustness. Here, those parameters were defined as described by the International Organization for Standardization (ISO) [30] and with regards to the flow cytometric application as proposed in a previous report [15].

(1) Specificity:

Due to the complexity of the assay, different parameters must be inspected. In order to show that the response of T cells was explicitly analysed, freshly isolated PBMCs were used and stained with fluorophore-conjugated antibodies according to the MLR protocol. Where indicated, CD3 BV421 (Cat# 344833, BioLegend, Amsterdam, The Netherlands) was used instead of VPD450, and CD14 PE (Cat# 367104, BioLegend, Amsterdam, The Netherlands) was used additionally. Using flow cytometry, the proportion of CD3^+^ T cells, CD5^+^ and CD4^+^ or CD8^+^ T cells, CD14^+^ monocytes and non-T cell lymphocytes was determined after 0–6 days of stimulation in vitro. Assuming a Gauss distribution, the proportion of T cells identified with different gating strategies was compared using a repeated measure one-way ANOVA at α = 0.05. Tukey adjusted *p*-values of <0.05 were considered significant.

In order to verify the specific inhibition of T cells by viable MSCs, a dilution series of different ratios of live MSCs to formalin fixed (dead) MSCs was tested. Therefore, MSCs were thawed and 100%, 70%, 50%, 30% or 0% of the final concentration were seeded in MSC medium as described above. After letting the cells adhere overnight, MSCs were carefully washed with PBS and fixed with 4% formaldehyde (Cat# 15512, Sigma-Aldrich, Taufkirchen, Germany). Then, 0%, 30%, 50%, 70% and 100% viable but irradiated MSCs in MLR co-culture medium were added to the respective wells. PBMCs were isolated, and MLRs were performed as described above. The limit of blank (LOB) was determined for stimulated PBMCs in the presence of 100% dead MSCs. In agreement with Wood et al. [15], the LOB was considered as the mean value + 1.645 standard deviations (SDs). The limit of detection (LOD) was then estimated as LOB + 1.645 SD.

(2) Accuracy:

A true value for the MLR was defined as the result of the BrdU-MLR assay, which has been used as the performance criterion of the MSC qualification prior to the development of the FACS-based MLR. Therefore, a parallel setting of the FACS-based MLR and BrdU-MLR was performed using MSCs from the same batch and PBMCs from the same donors in a 1:1 ratio. Concordance of the inhibition of proliferation was compared according to Pearson.

(3) Precision:

The repeatability, also known as intra-assay precision, was established by measuring the inhibition of T cell proliferation by MSCs for each experiment in triplicates and was then determined within each triplicate measurement for a total of 30 MLRs. The coefficient of variation (CV) was determined for each experiment, and the mean CV was defined as a validation parameter. For the intermediate precision, the Dil values of PBMC proliferation were analysed in the presence or absence of MSCs for a total of *n* = 30 MLRs. Here, the SD was defined as a validation parameter.

(4) Detection limit, (5) quantitation limit, (6) linearity and (7) range:

To find out how different PBMC:MSC ratios affect the proliferation of PBMCs, the inhibition of T cell proliferation was examined after co-culture in 1:1, 1:0.2, 1:0.1 and 1:0.05 ratios. The aim was to test how low the chosen MSC concentration can be in order to still detect their inhibitory potential on PBMC proliferation and to identify the dynamic range where the assay is discriminatory. The LOB and the LOD were determined for stimulated PBMCs in the absence of MSCs as described above. Additionally, the number of events was established, which was necessary to identify a peak as a generation. Therefore, the LOB for one generation was established by retrospectively analysing the number of events of VPD450-stained but unstimulated cells. The LOD was calculated as described above, and the lower limit of quantitation (LLOQ) was experimentally determined. PBMCs were stained and seeded according to the MLR protocol. After 2 days of CD3 and CD28 co-stimulation, stimulated and unstimulated PBMCs were mixed in defined ratios of 1:1, 1:3, 1:5, 1:10, 1:30, 1:50, 1:100, 1:300, 1:500 and 1:1000. The samples were then stained and measured according to the MLR protocol. The calculated value of proliferating cells was compared to the actual measured value and the LLOQ was considered as the lowest value above the LOD at which the CV of the triplicate measurement achieved <30% CV [15,32,33]. The “calculated value” was determined by multiplying the dilution factor by the “measured value” of the positive control. The correlation of the datasets was analysed using the non-parametric Spearman correlation coefficient.

In order to ascertain the linearity of the inhibitory effect of MSCs, different MSC concentrations were tested while keeping the number of PBMCs constant. The Dil values, the number of daughter generations and the inhibition of T cell proliferation were determined for PBMC:MSC ratios of 1:1, 1:0.2, 1:0.1 and 1:0. Obtained data were tested for departure from the normality distribution using a two-tailed Shapiro–Wilk test with α = 0.05 and was further analysed by computing Pearson’s correlation coefficients for a 95% confidence interval (α = 0.05).

(8) Robustness:

For *n* = 30 different PBMC donor pairs, the Dil values and the formation of daughter generations after co-stimulation for four days were analysed to assess the extent of biological inter-variability of T cell proliferation. Additionally, the respective inhibition by *n* = 30 MSC batches in a 1:1 ratio was calculated. The robustness was determined using a CV value < 15%.

## 3. Results

### 3.1. Assay Optimization Using VPD450 for Lymphocyte Proliferation Tracking

No changes in the viability of PBMCs were observed when the staining concentration of VPD450 was increased (Figure 1A). Hence, the maximal concentration of VPD450 was further used for high staining intensity without running off the axis. The viability of unstained and VPD450-stained PBMCs was compared for *n* = 12 experiments (Figure 1B), and no significant difference between matched donor pairs was found (*p* = 0.5693, two-tailed Wilcoxon test with α = 0.05). To define the ideal time point for the read-out, the aim was to generate a strong proliferative response with a VPD450 signal that was clearly distinguishable from the autofluorescence. Figure 1C shows the results of the time trial evaluating proliferation after one to six days. After two days, only one daughter generation had formed correlating to approximately 40% proliferation.

After five days, the proliferation signal starts to overlap with the autofluorescence of unstained cells. Hence, the best results were obtained after four days with a proliferative response (Dil) of around 90%. The effect of different MSC concentrations on T cell proliferation was evaluated. The proliferative response, assessed as a loss in the VPD450 staining intensity, was reduced in a dose-dependent manner in the presence of MSCs (Figure 1D–F) while showing a comparable signal between replicates.

### 3.2. Specificity

To define a parameter for the assay’s specificity, the intended use of the MSCs must be considered. As our MSCs are used for the treatment of GvHD, which is mainly induced by T cells, the specific inhibition of T cells by MSCs was investigated. However, after co-stimulation of PBMCs with monoclonal anti-CD3 and anti-CD28 antibodies, an impaired signal separation of CD3-positive and CD3-negative cells was observed in the presence of MSCs (Figure 2A).

To compensate for the impaired identification of CD3-positive cells, an alternative gating strategy was employed using the classical T cell marker CD5, which may also be expressed on B cells. CD3^+^ cells and T cells identified with CD45^+^/CD5^+^/CD4^+^ or CD8^+^ were enumerated, and the concordance of CD3 and CD5 expression was confirmed (Figure 2B–D). Further, the changes in PBMC subpopulations were analysed over time. In response to anti-CD3 and anti-CD28 co-stimulation, T cells overgrew the remaining mononuclear cells within a few days. Before the start of stimulation, mean values of 39.6% ± 7.6% T cells, 35.2% ± 1.3% other lymphocytes and 25.3% ± 9.0% monocytes were detected. On the following days, the T cell proportion increased, whereas the other lymphocytes and monocytes decreased. As of day four, more than 90% of the cells exhibited a T cell phenotype (Figure 2E).

To verify the specific inhibition of T cells by MSCs, a dilution series of different ratios of live MSCs to formalin-fixed (dead) MSCs was tested (Figure 3). Based on the inhibitory effect of 100% dead MSCs, a LOD of 35.7% was calculated for the 1:1 PBMC:MSC ratio, 21.6% for the 1:0.2 ratio and 10.4% for the 1:0.1 ratio with respective LOBs of 30.3%, 13.6% and 6.7%. Substituting viable MSCs with dead cells resulted in a significant reduction in MSC efficacy, although dead MSCs still exhibited modest inhibitory properties. Distinguishing between the effect of live and dead MSCs became increasingly difficult as the PBMC:MSC ratio decreased. At a concentration of 1:0.2, only 100% viable MSCs were able to induce the required inhibitory effect above the LOD. Further, at a 1:0.1 ratio, the inhibitory effect was merely above the LOB but did not exceed the LOD.

The same pattern was visible for the formation of T cell generations (Figure 3D–F), where the LOD calculated for *n* = 30 stimulated PBMCs in the absence of MSCs was applied (see Figure 4B). Hence, the results suggest a PBMC:MSC ratio of 1:0.2 as most sensitive to discriminate more functional from less functional batches.

### 3.3. Precision and Accuracy

In *n* = 30 MLRs, each experiment was performed in triplicates and at three different MSC concentrations. The intermediate precision was analysed using the spread of overall Dil values (Figure 5A) and the repeatability or intra-assay precision was analysed using the SD of Dil values within each triplicate measurement (Figure 5B). Thereby, a high intermediate precision for inter-assay comparison with overall SD values below 15% was observed for PBMCs alone or in the presence of MSCs (Figure 5A). High repeatability in the proliferation inhibition was found for the intra-assay comparison. Mean SD values of 2.0%, 1.5% and 1.3% and corresponding mean CV values below 10% with 9.2%, 2.7% and 1.5% for 1:1, 1:0.2 and 1:0.1 ratios were calculated, respectively (Figure 5B).

The accuracy of the FACS-based MLR was assessed by comparing the measured inhibition to the inhibition according to the BrdU-MLR as a reference method. The results of *n* = 10 experiments show a linear correlation (r = 0.9021, *p* = 0.0004) close to 1 (Figure 5C).

### 3.4. Detection Limit, Quantification Limit, Linearity and Range

The MLR is a functional rather than a phenotypic assay aiming to analyse the inhibitory effect of MSCs on T cell proliferation. In order to ascertain the dynamic range of the assay PBMC:MSC ratios of 1:1, 1:0.2, 1:0.1, 1:0.05 and 1:0 were tested. The LOB of 6.8% and the LOD of 13.6% were defined using the SD of the Dil values from *n* = 30 biological replicates of stimulated PBMC in the absence of MSC (1:0 ratio, blank control).

To define the range of the assay, *n* = 7 MLRs were performed with PBMC:MSC ratios of 1:0.05. Inhibition of 4.3% ± 7.2% was found, which is below the LOB and was no longer distinguishable from 0. For the 1:0.1 ratio, *n* = 30 experiments were analysed, and a mean inhibition of 14.5 ± 12.7% was observed. The inhibitory potential of MSCs was above 0, and the mean inhibition was above the LOD, therefore defining the lower LOD at the 1:0.1 ratio (Figure 4A). The upper end of the validated range was at the 1:1 ratio. The actual upper LOD was not determined since it was not relevant for the intended use of the assay. In terms of cell divisions, a LOB of 5.4 generations and a LOD of 4.2 generations were calculated. Accordingly, the 1:0.1 PBMC:MSC ratio sits exactly on the verge of the LOB, while the 1:0.2 and 1:1 ratios were above the LOD with 3.7 and 1.8 generations on average. Thus, a dynamic range to detect the inhibition of T cell proliferation by MSCs spanning from a PBMC:MSC ratio of 1:1 to 1:0.1 was validated. For the formation of daughter generations, a range from 1:1 to 1:0.2 was validated.

To define a peak as a daughter generation, a clear clustering of the population is required. The LOB was calculated as 62 ± 22 events and thus, the LOD was at 97 events. Subsequently, a dilution series was performed to calculate the respective LLOQ experimentally (Figure 4C). A rank correlation close to 1 (r = 0.9392, *p* < 0.0001) was found and CV values of ≤21% were obtained for calculated event numbers between 10 and 100. Therefore, the LLOQ for a peak was defined as 100 events.

T cell proliferation (Figure 6A), formation of daughter generations (Figure 6B) and the inhibition of T cell proliferation by MSCs (Figure 6C) was investigated to assess linearity in the MSC’s effect on T cells. The Dil values for T cell proliferation in co-culture with MSCs can be described by a linear correlation (r = −0.9532) with statistical significance (*p* = 0.0468). For emerging T cell generations in the presence of MSC, a linear correlation (r = −0.9147) was found but without statistical significance (*p* > 0.05). Lastly, T cell inhibition by MSCs was described using a linear correlation (r = 0.9538) with statistical significance (*p* = 0.0462).

### 3.5. Robustness

In *n* = 30 MLRs, the overall proliferation of T cells ranged from 78.9% to 98.1% with a mean value of 91.5% ± 4.2% (Figure 7A). As a parameter for the susceptibility to biological inter-variability of T cell proliferation, the CV value was determined, and a variation of 4.6% was found. The average number of T cell generation was 6.4 ± 0.6 and had a CV of 9.7% (Figure 7B). Additionally, the variation in the inhibition in the presence of MSCs in a 1:1 ratio was analysed. While the tested batches of MSCs exhibited a wide range in inhibitory potential from 50.9% to 96.0%, the CV value was relatively low at 13.0% (Figure 7C). This corresponds to a mean inhibition of 77.4% ± 10.0%.

## 4. Discussion

A FACS-based MLR assay was developed and formally validated to assess the pharmacological potency of MSCs intended as an immunotherapeutic approach to alleviate GvHD, i.e., their potential to suppress alloreactive T cells. The ICH Q2 guidelines as well as the recommendation for the validation of flow cytometric assays by O’Hara et al. and Wood et al. [15,34] provided the framework for the validation. Accordingly, the parameters specificity, precision, accuracy, detection limit, quantitation limit, range, linearity and robustness were considered. As per Lee et al. [35], our FACS data were classified as quasi-quantitative since the obtained results are numeric and are directly dependent on the sample characteristics but do not have a calibration standard [15,35,36].

While many different MLRs and T cell proliferation assays have been described in the literature [10,25,26,27,37,38,39], they appear to differ tremendously in terms of their basic setup conditions. For proliferation tracking of PBMCs, the detection methods include a stable isotope incorporation [10], spectrometric analysis of substrate conversion [39] or fluorescence-based detection techniques [23,24,25,37]. Radiolabelling or photo-spectrometric detection methods measure the response of all cells present in the culture during the read-out within a pre-defined time window (end-point assay) instead of a specific target population. Although those assays are reasonably robust for the evaluation of PBMC proliferation, our flow cytometry-based MLR provides the additional benefit of specifically measuring T cell proliferation during the total incubation time and allows for the assessment of viability and the discrimination between effector and target cells. FACS-based proliferation assays oftentimes introduce covalently binding proliferation dyes such as originally CFSE/CFDA (carboxyfluorescein succinimidyl ester/carboxy-fluoresceindiacetate-acetoxymethylester) [25,26,40] or more recently violet proliferation dyes [27,41], which progressively halve fluorescence intensity during cell division. Although many groups including ourselves have used CFSE/CFDA dye dilution to measure T cell proliferation, even for pharmaceutical purposes [42,43], CFSE has been described as cytotoxic and losing a lot of its fluorescence intensity within one day of culture [28,44]. For VPD450, only a minor loss in the fluorescence intensity was found and no cytotoxic effect for up to 9 µM staining concentration tested was observed. We thereby avoided an additional working day for measuring the initial signal intensity of undivided cells on day one. A higher initial staining intensity further permits the detection of additional daughter generations before the signal overlaps with the cells’ autofluorescence. Our results are in agreement with others [45] who directly compared the viability of PBMCs after staining with either CFSE or VPD450. Here, increasing concentrations of CFSE staining led to a decline in viability, while fewer than 20% dead cells were found 96 h after staining with VPD450 regardless of the concentration. Additionally, staining with VPD450 resulted in better peak separation than CFSE. In contrast to Ten Brinke et al. [45], where decreased responsiveness of T cells was described as a consequence of increasing staining concentrations of CFSE and VPD450, no impaired cell proliferation was observed in our optimized assay setup.

To define the optimal time point for the read-out of our FACS-based MLR, T cell proliferation was examined over seven days. Day four was optimal in terms of a strong proliferative response, which was also distinguishable from autofluorescence. Four days of culture have often been described for MLR assays when a strong activation stimulus was provided [10,25,26,37,39].

The most physiological way to model T cell activation in the context of GvHD occurs during the co-culture of two HLA-mismatched PBMC donors resulting in T cell activation and proliferation in response to the mismatch. However, mere co-culture of two PBMC donors does not always result in robust T cell activation with a strong proliferative response after four days [26]. In vivo, predominantly alloreactive T cells are activated and induce GvHD pathology [16]. Artificially enhancing the T cell response differs from the physiological process, since T cell receptors on all T cells are targeted including non-alloreactive cells. The effect of MSCs on the immune system strongly depends on an inflammatory microenvironment [39,46,47]. Hence, a deviation from the physiological process is needed to achieve the required extent of T cell activation. Ketterl et al. [26] co-cultured a pool of ten different PBMC donors to increase the allogeneic stimulus. Nevertheless, seven days of PBMC culture were needed to obtain a sufficient alloresponse. Another solution to enhance the reaction in vitro is the co-culture of dendritic cells (DCs) and T cells. However, it is crucial to co-culture those cells in the correct ratio [48,49,50,51,52]. Alternatively, T cell activation can be induced using artificial stimuli. Unspecific activators include phytohaemagglutinin (PHA) [38,39,53,54], concanavalin A (ConA) [10,53,55], bacterial toxins [37,56] or monoclonal antibodies [25,26]. Activation of T cells via the CD3 and CD28 receptors leads to the proliferation and secretion of pro-inflammatory cytokines [18,57,58]. To detect a reduction in proliferation in the first place, and to allow the MSCs to exert their immunoregulatory effect, successful T cell stimulation is required. Hence, to determine the antiproliferative effect of MSCs using our MLR, the allogeneic reaction was enhanced by adding purified anti-CD3 and anti-CD28 monoclonal antibodies to the culture, which simulate the presence of the first and second T cell signal. Thereby, specifically, T cell activation receptors are engaged whereas HLA-disparate antigens are present by the co-culture of biologically different donors. This technique allows the fastest implementation and requires little handling time on the day of seeding.

After stimulation with anti-CD3 antibodies, a drastic decrease in CD3 staining intensity occurred. We observed overlapping populations of CD3-positive and -negative cells. Therefore, an alternative gating strategy was employed using the CD5 marker, which is expressed on human T lymphocytes and B lymphocytes [59,60]. T cells were identified by the co-expression of CD5 and CD4 or CD5 and CD8. Neither CD4 nor CD8 are expressed by B cells but both are used to distinguish different T cell subtypes [61,62]. Gating of T cells that co-express CD5 and CD4 or CD8 allowed specific T cell identification, which did not differ significantly from CD3-positive T cells.

In order to show that the measured T cell inhibition was specifically caused by viable MSCs, decreasing proportions of live MSCs mixed with formalin-fixed cells were used. The presence of 100% dead cells still reduced T cell proliferation by 24.2% for the 1:1 ratio and to a reduced extent for lower MSC numbers. Multiple groups described that the effect of MSCs is to some extent caused by soluble modulators [38,63,64]. Moreover, there are attempts to use cell-free MSC therapies administering secreted molecules only to ensure a higher clinical safety [65]. Both findings suggest that viable and metabolically active MSCs are required to reduce T cell proliferation. On the other hand, MSCs express the immunoregulatory molecules programmed death ligand (PD-L) 1 and PD-L2 on their surface, which can reduce the T cell response [63]. Formaldehyde-fixing of the cells prevents the release of soluble molecules, which reduces the functionality of the MSCs. In addition, by fixing the cells, the surface molecules are cross-linked, which could impair their function as well [66]. It has further been suggested that apoptotic rather than viable MSCs reduce GvHD in mouse models [67]. This might explain our observation that even fixed MSCs inhibited T cell proliferation to some degree. The aim of this co-culture approach was to test whether the MLR can distinguish more functional (100% viable) MSC batches from less functional ones (<70% viable). A strong decline in anti-proliferative function was observed when mostly dead MSCs were used. Interestingly, for the 1:1 ratio, T cell inhibition was above the LOD for up to 50% of dead cells. The decline in MSC quality was more pronounced for the lower PBMC:MSC ratios, where only the 100% viable cells caused an inhibition above the LOB. Hence, we recommend analysing the 1:0.2 and 1:0.1 ratios additionally and setting limits for each to ensure the clinical application of high-quality MSC batches only.

O’Hara et al. [34] investigated the required precision thresholds for the validation of different assay types. For methods approved by the American Food and Drug Administration (FDA), CV values of <10% should be obtained within one experiment and CV values <15% for inter-assay comparison. Our MLR meets both requirements for all PBMC:MSC ratios tested, as values below 10% for the repeatability (intra-assay comparison) as well as <15% for the intermediate precision (inter-assay comparison) were observed.

As specified in the ICH guideline, five different PBMC:MSC ratios were tested, including the positive control without MSCs. While the 1:1 ratio yielded the strongest inhibition, it is important to additionally consider the inhibitory potential of lower cell numbers for improved sensitivity. Most publications regarding the immunomodulatory potential of MSCs have also addressed the range in which they are effective [10,25,26,27,37,38]. However, to our knowledge the analytical detection limit has not been calculated in either of those approaches. Considering the calculated LOD, a range of 1:1 to 1:0.1 was determined for the assay, which is a rather narrow range of the MLR. Nicotra et al. and Le Blanc et al. [10,27] demonstrated a range of 1:1 to 1:0.01 with slightly to no inhibition at lower concentrations. The difference in range might be influenced by using monoclonal antibodies to enhance the alloreaction in our experiments. It is likely that MSCs reduce T cell proliferation more efficiently for a moderate rather than an overshooting response caused by an artificial stimulus. Other groups who induced T cell proliferation with mitogens or monoclonal antibodies used similar PBMC:MSC ratios as we have [25,26,37]. Di Nicola et al. [38] showed that MSCs can inhibit T cell proliferation induced with DC-mediated activation as well as with PHA stimulation. However, the extent of inhibition measured seemed to be less pronounced after using PHA as the stronger stimulus.

The linearity of a quasi-quantitative assay is technically not required for its validation [15,34]. Yet we tested for linear correlations in proliferation, formation of daughter generations and inhibition of proliferation. Statistically significant linear correlations were found for T cell proliferation and inhibition by MSCs, but not for the formation of daughter generations. Although correlation coefficients close to 1 were determined, it is likely that inhibition in the 1:1 ratio is at a saturated state.

The minimal number of events required to define a cluster of events as a daughter generation was defined as 100 events. Above this threshold, the number of measured events was highly accurate. However, in terms of linearity and precision, the number of daughter generations failed to reach the acceptance criteria. Hence, it does not seem to be appropriate for the assessment of MSC inhibitory potential. We, therefore, propose that the number of generations should be considered as internal control only. Moreover, we suggest specifying the detection of a minimum of five daughter generations for the stimulated PBMC control in the absence of MSCs (1:0 ratio) as an acceptance criterion for the assay. This indicates sufficiently strong T cell proliferation allowing the analysis of the MSCs’ anti-inflammatory effect. The use of daughter generations as an acceptance criterion further entails the verification of successful VPD450 staining by allowing the identification of distinguishable peaks.

Monoclonal anti-CD3 and anti-CD28 antibodies were used to enhance the T cell response and thereby compensate for biological differences in terms of the abundance and response intensity of alloreactive T cells. For 30 different PBMC donor pairs high and highly consistent proliferative responses with high precision were observed. Thus, an additional thorough validation process of multiple donor PBMC pools as proposed elsewhere [26,27] is not required for our method. The combination of different biological donors and simultaneously inducing a robust T cell response has further advantages. Firstly, only a strong T cell response allows a reliable and differentiated analysis of MSC potency and thereby the identification of products with impaired anti-proliferative effects. Secondly, the T cell response is accelerated, and the effect of MSCs can be analysed quickly and with little handling time. The FACS-based MLR assay was validated using three different MSC banks, consisting of MSCs generated from a pool of bone marrow mononuclear cells of eight donors each. The inhibitory potential of the 30 batches tested for 30 different donor pairs shows precise and reproducible results. A CV of 13.2% was observed, which meets the acceptance criteria of the FDA for inter-assay precision and highlights the robustness of the system [34,68,69].

While MSCs should be most effective at the 1:1 ratio, the assessment of MSCs in lower concentrations was required to detect qualitative differences in MSC viability. We suggest that for the release of an MSC batch, T cell inhibition values of 40% and above must be obtained in the 1:1 ratio. We further propose to include the analysis of the 1:0.2 and 1:0.1 ratios to detect inferior MSC batches. Each condition should be prepared in technical triplicates and the effect of random MSC products of each production batch should be analysed for three different donor pairs. In any case, it is difficult to predict effectiveness using a single method only. The International Society for Cell Therapy proposed the development of a matrix approach to accurately determine the potency of MSCs to improve the efficacy and safety for clinical applications [70]. Our assay holds the potential to be implemented as one of the multiple performance criteria for the release of clinical MSC products, as the staining panel can easily be expanded with activation markers, markers for T cell subpopulations or combined with RNA and secretome analysis.

## 5. Conclusions

In the current study, staining of PBMCs with VPD450 allowed precise T cell proliferation tracking. In contrast to CFSE, we found superior peak separation using VPD450 and no changes in PBMC viability and responsiveness. Therefore, our MLR setup enables the assessment of MSCs’ inhibitory effect specifically on T cell proliferation with high accuracy and precision for PBMC:MSC ratios of down to 1:0.1. It thereby allows the identification of MSC batches with strong antiproliferative properties compared to less potent ones. In conclusion, we propose to use this MLR assay as a surrogate potency assay for the quality assessment of clinical MSC batches.

## Figures and Tables

**Figure 1 cells-12-00850-f001:**
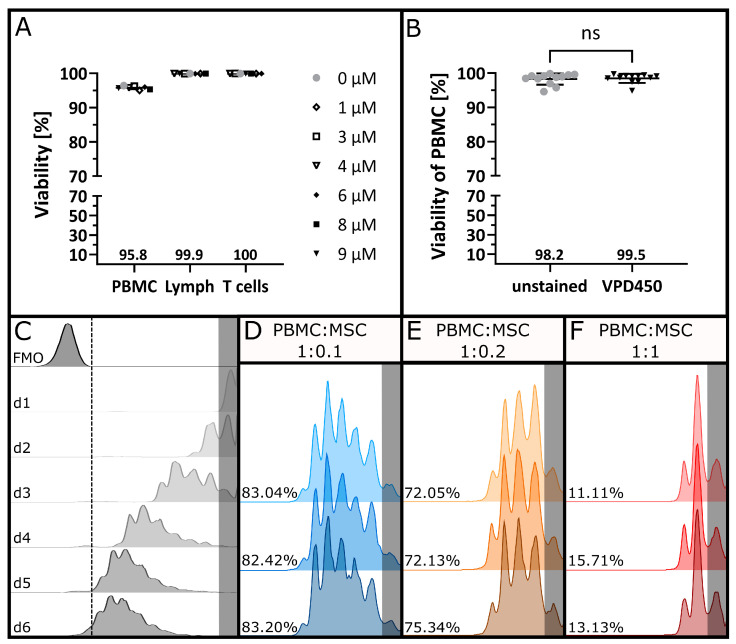
Proliferation tracking of PBMCs. (**A**) Viability of PBMCs was determined using 7AAD after staining with increasing concentrations of VPD450 to assess the cytotoxicity of the proliferation dye. Mean values are indicated below the plots. (**B**) Viability of PBMCs after staining with VPD450 was compared to the viability of the respective unstained controls. Mean values for *n* = 12 stainings are indicated below the plots. Statistical analysis was performed using a two-tailed Wilcoxon test with α = 0.05. (**C**) VPD450-based proliferation tracking of PBMC donor pairs after co-stimulation with anti-CD3 and anti-CD28 antibodies after one, two, three, four, five and six days in culture. Histograms display VPD450 fluorescence intensity after gating on CD5^+^ and CD4^+^ or CD8^+^ T cells. (**D**) Triplicate measurement of co-stimulated PBMC donor pairs after four days of co-culture with MSCs in a 1:0.1 ratio, (**E**) in a 1:0.2 ratio and (**F**) in a 1:1 ratio. Undivided generations are highlighted in grey, and respective Dil [%] values are indicated for each replicate.

**Figure 2 cells-12-00850-f002:**
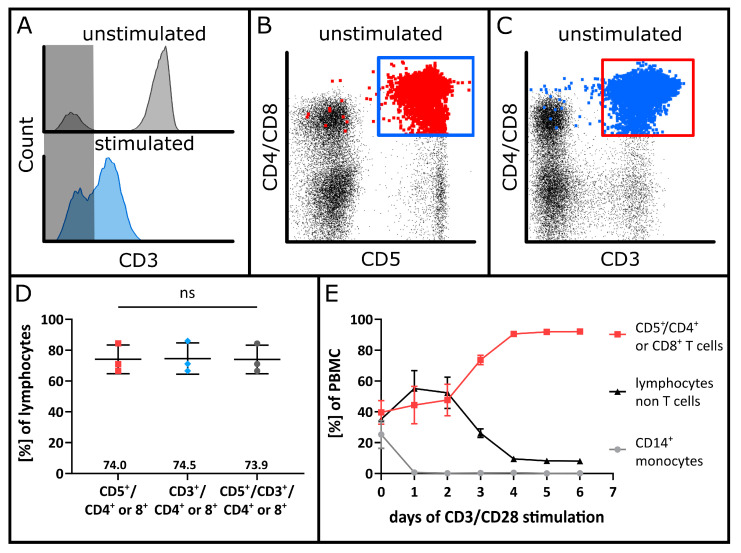
Specific analysis of T cells as target population. (**A**) Loss in the distinct separation of T cells using cell surface staining of the CD3 receptor after co-stimulation with anti-CD3 and anti-CD28 antibodies. The CD3 gate was set according to the unstimulated control visible in the upper histogram. The area containing CD3 negative cells is highlighted in grey. PBMCs after co-stimulation in the presence of MSCs are depicted in the lower histogram. The CD3^−^ and CD3^+^ populations are overlapping, which impedes the specific identification of T cells. (**B**) Representative dot plot of PBMCs prior to MLR co-cultures gated on CD3^+^ and CD4^+^ or CD8^+^ for T cell identification. T cells alternatively identified with CD5^+^ and CD4^+^ or CD8^+^ are displayed in red and emphasised. (**C**) Representative dot plot of CD5^+^ and CD4^+^ or CD8^+^ T cells prior to seeding for MLR co-cultures. CD3^+^ and CD4^+^ or CD8^+^ T cells are displayed in blue and emphasised. (**D**) Comparison of unstimulated T cell populations identified with CD5 or CD3 combined with CD4^+^ or CD8^+^ surface marker expression and co-expression. Mean values are indicated above the *X*-axis. Statistical analysis was performed using a repeated-measure one-way ANOVA with α = 0.05. (**E**) Specific enrichment of T cells after co-stimulation with anti-CD3 and anti-CD28 antibodies. Changes in PBMC subpopulations are shown on the day of seeding and additionally after one, two, three, four, five and six days of culture. T cells are defined as CD5^+^ and CD4^+^ or CD8^+^ cells.

**Figure 3 cells-12-00850-f003:**
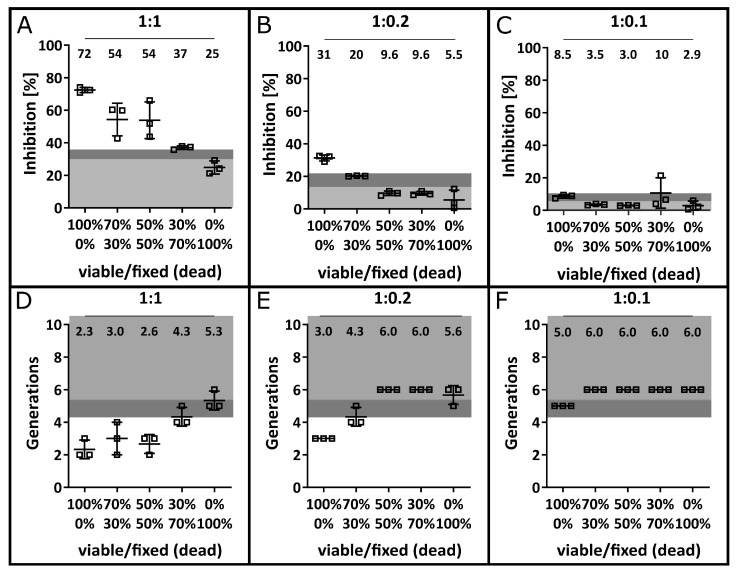
Specific inhibition of T cell proliferation by viable MSCs. MSCs were made non-functional with formalin fixation, mixed with vital MSCs at the indicated viable/fixed (dead) ratios and used in MLR co-cultures. The respective mean values of *n* = 3 independent experiments are indicated within each plot. Areas highlighted in light and medium grey mark the calculated LOD and LOB based on 100% dead MSCs in the respective concentration. The inhibition of T cell proliferation was determined in the presence of MSCs after co-culture in a 1:1 ratio (**A**), in a 1:0.2 ratio (**B**) and in a 1:0.1 ratio (**C**). Additionally, the number of T cell generations is displayed after co-culture with MSCs in a 1:1 ratio (**D**), a 1:0.2 ratio (**E**) and a 1:0.1 ratio (**F**) with the LOB and LOD defined by Figure 4.

**Figure 4 cells-12-00850-f004:**
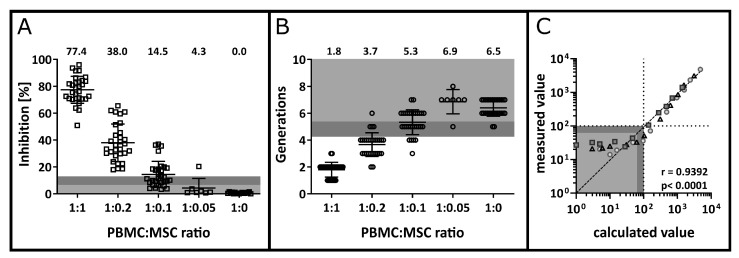
Detection limit and quantitation limit. (**A**) Dose-dependent inhibition of T cell proliferation by MSCs. PBMCs and MSCs were co-cultured in different ratios to define the lowest MSC concentration for which the inhibition of proliferation can still be detected. Based on the 1:0 ratio, a LOB of 6.8% and a LOD of 13.6% were determined and are highlighted by medium and light grey areas. Mean values for *n* = 30 (1:1, 1:0.2, 1:0.1, 1:0) or *n* = 7 (1:0.05) are indicated above the plot. (**B**) Number of detectable T cell generations with respect to MSC co-incubation at different ratios. The LOB at 5.4 generations and the LOD at 4.3 generations were calculated based on T cell generations in the absence of MSCs and are highlighted within the plot in light and medium grey (*n* = 30 and *n* = 7 for 1:0.05). (**C**) Determination of the LLOQ for one daughter generation. Anti-CD3 and anti-CD28 co-stimulated cells were spiked into unstimulated cells as a negative control for an undivided generation. A dilution series was performed, and the calculated value of divided PBMCs was plotted against the number of detected events (measured value). Data were obtained in three independent experiments. Circles, squares and triangles represent data obtained within one titration experiment, each. Values below the LOD of 97 events and the LOB of 62 events are highlighted in light and medium grey, respectively, and the LLOQ of 100 events is indicated by the dotted lines. The dataset was analysed by computing the Spearman correlation coefficient (r = 0.9392, *p* < 0.0001).

**Figure 5 cells-12-00850-f005:**
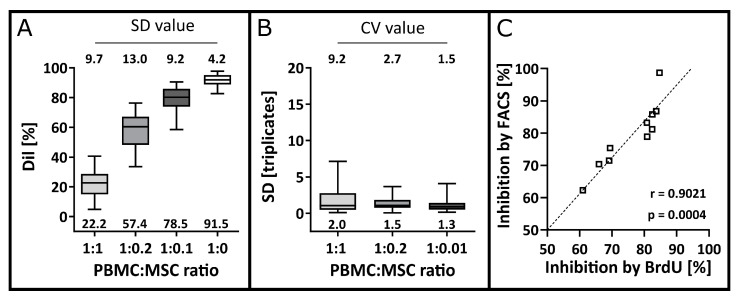
Precision and accuracy. (**A**) The intermediate precision was determined by plotting the mean Dil values of each triplicate for *n* = 30 MLRs. Overall mean values are indicated below the box plots of the 90% confidence interval. The overall SD values were determined as validation parameters and are indicated above the box plots. (**B**) The repeatability or intra-assay precision was determined by analysing the SD for each triplicate measurement for each PBMC:MSC ratio. Mean SD values are indicated below the box plots of the 90% confidence interval. The CV was analysed for each triplicate as a validation parameter, and the respective mean values are indicated above the box plots (*n* = 30). (**C**) Accuracy of T cell inhibition using the FACS-based MLR in comparison to the BrdU-MLR. Inhibition values measured with FACS and with BrdU were plotted against each other. The degree of linear correlation was computed using the Pearson correlation coefficient with α = 0.05 resulting in r = 0.9021 and *p* = 0.0004.

**Figure 6 cells-12-00850-f006:**
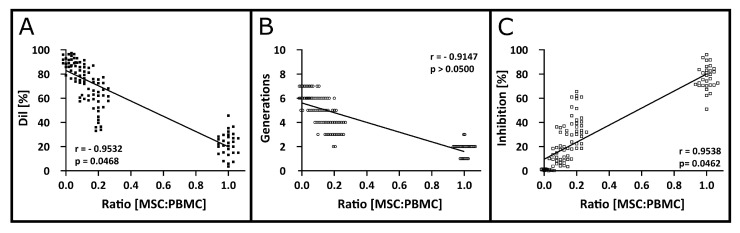
Linearity in T proliferation after co-culture in different PBMC:MSC ratios. (**A**) Linear correlation of T cell proliferation shown by the detected Dil values in the presence or absence of MSCs. The Pearson correlation coefficient was r = −0.9532 with statistical significance (*p* = 0.0468). (**B**) Formation of T cell generations after co-stimulation of PBMCs in the presence or absence of MSCs. The Pearson correlation coefficient was r = −0.9147 and *p* was >0.05. (**C**) Linearity in the calculated inhibition of T cell proliferation by MSCs using the Dil as evaluation parameter with r = 0.9538 and *p* = 0.0462. The degree of linear correlation was computed using the Pearson correlation coefficient with α = 0.05.

**Figure 7 cells-12-00850-f007:**
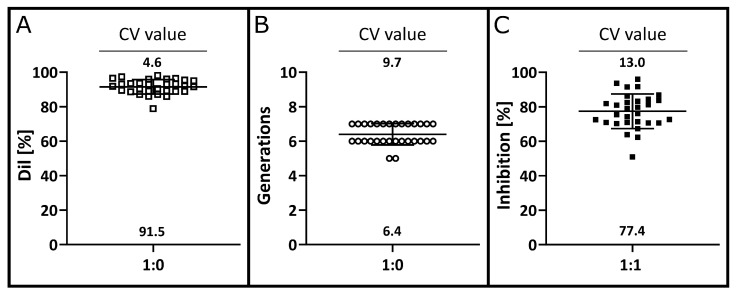
Robustness in the proliferative response of different PBMC donor pairs. After four days of T cell stimulation, the proliferative capacity using the Dil value (**A**), the number of T cell generations (**B**) and the inhibition by MSC in a 1:1 ratio (**C**) was determined for *n* = 30 PBMC donor pairs. The mean value of each analysis parameter is indicated at the bottom of the plot. The CV values were chosen as validation parameter and are indicated above the plots.

## Data Availability

Not applicable.

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
