# Peer review of "Validation of an ICH Q2 Compliant Flow Cytometry-Based Assay for the Assessment of the Inhibitory Potential of Mesenchymal Stromal Cells on T Cell Proliferation"

_cells, 2023, doi:10.3390/cells12060850_

Round 1
Reviewer 1 Report
Dear authors, this is an excellent, very interesting and timely article on an important topic, to better standardize the assessment of MSC immune-inhibitory potential on T-cell proliferation, by employing an ICH-Q2 compliant FACS assay based on the VDP450 dye PBMC proliferation staining (Validation described in detail in section 2.6, including 1) Specificity, 2) Accuracy, 3) Precision, 4) Detection Limit, 5) Quantitation Limit, 6) Linearity, 7) Range, and 8) Robustness). I highly appreciate this important work from your group, which will bring the whole field further forward and may also serve as a good template for other groups to establish this important method in a proper/reproducible fashion in their own labs. These types of studies should be performed more often. There are some very minor points that should be clarified / strengthened that I have included below, but altogether this is an excellent study.
Intro and Methods:
1) Page 2, line 46-47, considering efficacy and safety profile of MSC products (Ref 12+13), please consider including PMID 30711482 Trends in Molecular Medicine 2019 ”Intravascular MSC Therapy Product Diversification: Time for New Clinical Guidelines”, PMID 35641163 Stem Cells Translational Medicine 2022 “Improved MSC minimal criteria to maximize patient safety: a call to embrace tissue factor and hemocompatibility assessment of MSC products.”, PMID 35502223 Current Stem Cell Reports 2022 “Impact of Cryopreservation and Freeze-Thawing on Therapeutic Properties of Mesenchymal Stromal/Stem Cells and Other Common Cellular Therapeutics”. These references would also fit perfectly to further support this statement, considering the impact of donor heterogeneity, tissue origin, and manufacturing progress but also cryopreservation and freeze-thawing impact for safety and efficacy of MSC products.
2) Page 2, line 51-52 and 80-83: considering the mentioning of the ICH Q2 compliant potency assay, please cite some suitable references/webpage at this stage, where this is described in more detail for readers. Later on the authors refer to references number 29-31 in section 2.6, which could already be cited at this earlier stage to better inform the readers on this point.
3) Page 3, line 100 and line 124-139: considering VDP450 staining, please mention here that you performed a titration of the concentration of the dye on different cell types (shown in Figure 1A). Just wondering, did you consider doing a multi-center external validation study of all these assays for testing their reproducibility on independent flow cytometry facilities (not just in-house)?
4) Results: Experimental execution, to follow testing requirements for ICH-Q2 is very well done, as briefly summed up and commented on in the four respective sections below.
Section 3.1 and Figure 1 Well done Assay Optimization (Dye Titration, PBMC Viability, VPD450-labeled PBMC donor pair tracking, Triplicate Measurements of different MSC:PBMC ratios)
Section 3.2 and Figure 2+3 Both very interesting results, e.g. on different mixtures/ratios of life and PFA-fixed MSCs (specific inhibition by viable MSCs, Fig 3), but also on FACS staining with CD3 vs. CD5 staining after CD3/CD28-stimulation with slightly reduced CD3 expression (Fig 2).
Section 3.3 and Figure 4, once more very important analysis on precision & accuracy of the assay, to calculate SD and CV value and linear correlation
Section 3.4+3.5 and Figure 5+6+7, on detection limit, quantification limit, linearity and range, and robustness. Once more very well done, I have nothing to add here.
5) Considering the biological/scientific interpretation, in addition to testing BM-MSCs (n=30 MLRs), it would also be of interest to evaluate/test different types of MSCs from different tissue sources in this assay against the same set of PBMCs (only for the optimized conditions) and to compare them for their inhibitory effect, as compared by Ketterl et al. 2015 (Ref 24).
6+7) Discussion and Conclusion sections, very well done. Maybe strengthen in the conclusion the similarities and differences of VDB450- to CFSE-staining.
Author Response
Reviewer 1
Comments and Suggestions for Authors
Dear Reviewer,
We greatly appreciate your constructive critique and valuable suggestions.
Point-by-point response to the reviewers’ comments
Dear authors, this is an excellent, very interesting and timely article on an important topic, to better standardize the assessment of MSC immune-inhibitory potential on T-cell proliferation, by employing an ICH-Q2 compliant FACS assay based on the VDP450 dye PBMC proliferation staining (Validation described in detail in section 2.6, including 1) Specificity, 2) Accuracy, 3) Precision, 4) Detection Limit, 5) Quantitation Limit, 6) Linearity, 7) Range, and 8) Robustness). I highly appreciate this important work from your group, which will bring the whole field further forward and may also serve as a good template for other groups to establish this important method in a proper/reproducible fashion in their own labs. These types of studies should be performed more often. There are some very minor points that should be clarified / strengthened that I have included below, but altogether this is an excellent study.
Intro and Methods:
1) Page 2, line 46-47, considering efficacy and safety profile of MSC products (Ref 12+13), please consider including PMID 30711482 Trends in Molecular Medicine 2019 ”Intravascular MSC Therapy Product Diversification: Time for New Clinical Guidelines”, PMID 35641163 Stem Cells Translational Medicine 2022 “Improved MSC minimal criteria to maximize patient safety: a call to embrace tissue factor and hemocompatibility assessment of MSC products.”, PMID 35502223 Current Stem Cell Reports 2022 “Impact of Cryopreservation and Freeze-Thawing on Therapeutic Properties of Mesenchymal Stromal/Stem Cells and Other Common Cellular Therapeutics”. These references would also fit perfectly to further support this statement, considering the impact of donor heterogeneity, tissue origin, and manufacturing progress but also cryopreservation and freeze-thawing impact for safety and efficacy of MSC products.
Response:
According to your suggestion we replaced previous references 12 & 13 with 2 new references as following (line 47):
- Moll, G.; Ankrum, J.A.; Kamhieh-Milz, J.; Bieback, K.; Ringdén, O.; Volk, H.-D.; Geissler, S.; Reinke, P. Intravascular Mesenchymal Stromal/Stem Cell Therapy Product Diversification: Time for New Clinical Guidelines. Trends Mol. Med. 2019, 25, 149–163, doi:10.1016/j.molmed.2018.12.006.
- Cottle, C.; Porter, A.P.; Lipat, A.; Turner-Lyles, C.; Nguyen, J.; Moll, G.; Chinnadurai, R. Impact of Cryopreservation and Freeze-Thawing on Therapeutic Properties of Mesenchymal Stromal/Stem Cells and Other Common Cellular Therapeutics. Curr. Stem Cell Rep. 2022, 8, 72–92, doi:10.1007/s40778-022-00212-1.
2) Page 2, line 51-52 and 80-83: considering the mentioning of the ICH Q2 compliant potency assay, please cite some suitable references/webpage at this stage, where this is described in more detail for readers. Later on the authors refer to references number 29-31 in section 2.6, which could already be cited at this earlier stage to better inform the readers on this point.
Response:
According to your suggestion, the references were also cited on page 2. The respective references are now numbers as 14, 15 (line 54) and 30 (line 81).
3) Page 3, line 100 and line 124-139: considering VDP450 staining, please mention here that you performed a titration of the concentration of the dye on different cell types (shown in Figure 1A). Just wondering, did you consider doing a multi-center external validation study of all these assays for testing their reproducibility on independent flow cytometry facilities (not just in-house)?
Response:
- According to your suggestion we added on page 3 line 98 – 100 the following sentence:
“To find out the optimal dye concentration we performed a titration of VPD450, which was reconstituted according to the manufacturer instructions and added to the PBMC donor pairs.”
- Thank you very much for this suggestion. You are absolutely right that a multi-center external validation study is useful. So far, we have not tested the reproducibility on independent flow cytometry facilities. However, in the near future we plan to expand the capacity by a tech transfer to additional QC labs. The regulatory requirement is currently to test selected batches. An expansion to every batch produced is foreseeable, so capacity will need to be expanded. As soon as more laboratories perform the assay, intra-laboratory testing will be performed.
4) Results: Experimental execution, to follow testing requirements for ICH-Q2 is very well done, as briefly summed up and commented on in the four respective sections below.
Section 3.1 and Figure 1 Well done Assay Optimization (Dye Titration, PBMC Viability, VPD450-labeled PBMC donor pair tracking, Triplicate Measurements of different MSC:PBMC ratios)
Section 3.2 and Figure 2+3 Both very interesting results, e.g. on different mixtures/ratios of life and PFA-fixed MSCs (specific inhibition by viable MSCs, Fig 3), but also on FACS staining with CD3 vs. CD5 staining after CD3/CD28-stimulation with slightly reduced CD3 expression (Fig 2).
Section 3.3 and Figure 4, once more very important analysis on precision & accuracy of the assay, to calculate SD and CV value and linear correlation
Section 3.4+3.5 and Figure 5+6+7, on detection limit, quantification limit, linearity and range, and robustness. Once more very well done, I have nothing to add here.
5) Considering the biological/scientific interpretation, in addition to testing BM-MSCs (n=30 MLRs), it would also be of interest to evaluate/test different types of MSCs from different tissue sources in this assay against the same set of PBMCs (only for the optimized conditions) and to compare them for their inhibitory effect, as compared by Ketterl et al. 2015 (Ref 24).
Response:
The next step of this assay would be testing of these MSC against a pool of many more random PBMC-donors (ten or more), which ensures a multidirectional alloresponse as reported by Ketterl et al. (Ref.:24). We are also very curious whether these clinically grade MSCs, which were generated from pooled bone marrow mononuclear cells of eight healthy adult donors, would be as effective as against two PBMC donors. In addition, the allosuppressive potential of these MSCs will be compared to that of adipocyte-derived MSCs, which are available in our lab. We hope, in a near future to be able to have the first results concerning this interesting issue.
6+7) Discussion and Conclusion sections, very well done. Maybe strengthen in the conclusion the similarities and differences of VDB450- to CFSE-staining.
Response:
We strengthened the advantages of VPD450 over CFSE for our assay set-up in the discussion and added the following paragraph (line 488 – 451):
“We thereby avoided an additional working day of measuring the initial signal intensity of undivided cells on day one. A higher initial staining intensity further permits the detection of additional daughter generations before the signal overlaps with the cells’ autofluorescence.”
According to your suggestion we changed the Conclusion section as follows:
“In the current study, staining of PBMCs with VPD450 allowed precise T cell proliferation tracking. In contrast to CFSE, we found superior peak separation using VPD450 and no changes in PBMC viability and responsiveness. Therefore, our MLR setup enables the assessment of MSCs’ inhibitory effect specifically on T cell proliferation with high accuracy and precision for PBMC:MSC ratios of down to 1:0.1. It thereby allows the identification of MSC batches with strong antiproliferative properties compared to less potent ones. In conclusion, we propose to use this MLR assay as a surrogate potency assay for quality assessment of clinical MSC batches.”
Submission Date
07 February 2023
Date of this review
16 Feb 2023 08:43:54
Reviewer 2 Report
General Comments:
This is a solid contribution that can be useful to the field as MLR assays are utilized frequently for clinical MSC screening.
Specific comments:
Methods; section 2.6., (1), second sentence: It does not look like CD105 was used in any of the experiments. Please remove here and elsewhere.
Same section, second paragraph: Wouldn't heating or irradiation be better than formalin as that cross-links surface proteins and, thus, disrupts their function which could still be intact in dead cells. This might deserve a brief explanation.
Results; 3.2. Specificity, second sentence: This interpretation may not be correct. Couldn't diminution of CD3 signal be due to epitope blocking by stimulating antibody? This might deserve a brief explanation.
Same section, paragraph 2: Suggest showing directly that CD5 enhances differentiating CD4+ and CD8+ T cell subsets from other Leukocytes. Concordance between CD3 and CD5 could simply rfelect a lack of differentiating power between the two epitopes.
Author Response
Reviewer 2
Comments and Suggestions for Authors
Dear Reviewer,
We greatly appreciate your constructive critique and valuable suggestions.
Point-by-point response to the reviewers’ comments
General Comments:
This is a solid contribution that can be useful to the field as MLR assays are utilized frequently for clinical MSC screening.
Specific comments:
Methods; section 2.6., (1), second sentence: It does not look like CD105 was used in any of the experiments. Please remove here and elsewhere.
Response:
Thank you for the comment, we excluded CD105 from those sentences:
Line 126 – 129:
“Therefore, cells were pipetted through a 40 µm cell strainer (Cat# 43-10040-70, pluriSelect) and stained with CD45 FITC (Cat# A07782, Beckman Coulter), 7-AAD (Cat# A07704, Beckman Coulter), CD4 PC7 (Cat# 737660, Beckman Coulter), CD8 PC7 (Cat# 737661, Beckman Coulter) and CD5 APC (Cat# 345783, BD).”
Line 134 – 135:
“In order to determine the extent of T cell proliferation, a gating strategy was established for CD45+/7AAD-/CD5+/CD4+ or CD8+ singlet cells.”
Line 166 – 167:
“Where indicated, CD3 BV421 (Cat# 344833, BioLegend) was used instead of VPD450, and CD14 PE (Cat# 367104, BioLegend) was used additionally.”
Same section, second paragraph: Wouldn't heating or irradiation be better than formalin as that cross-links surface proteins and, thus, disrupts their function which could still be intact in dead cells. This might deserve a brief explanation.
Response:
As stated in the discussion, we assume that most of the allosupressive potential of MSCs is mediated by soluble molecules. Thus, we decided to use formalin as fixation agent, because this allows retention of soluble mediators within the cells, in contrast to heating, which may have destroyed the MSCs and consequently may cause the release of these molecules into MLR-supernatants. In addition, we used irradiated MSCs, which do not divide, but still may have the ability to produce and release mediators, which are responsible for the suppression of alloreaction in MLR, such as PGE2, IDO etc. Thereof, we prepared defined ratios of MSCs, which release suppressive molecules (viable and metabolically active MSCs) and less/ non-functional ones, which cannot release any of those proteins (fixed/dead MSCs).
According to your suggestion, we added a brief passage in the discussion and the respective reference as follows (line: 507 – 510):
“Formaldehyde-fixing of the cells prevents the release of soluble molecules, which reduces the functionality of the MSCs. In addition, by fixing the cells the surface molecules are cross-linked, which could impair their function as well [66].”
- Sompuram, S.R.; Vani, K.; Messana, E.; Bogen, S.A. A Molecular Mechanism of Formalin Fixation and Antigen Retrieval. Am. J. Clin. Pathol. 2004, 121, 190–199, doi:10.1309/BRN7-CTX1-E84N-WWPL.
Results; 3.2. Specificity, second sentence: This interpretation may not be correct. Couldn't diminution of CD3 signal be due to epitope blocking by stimulating antibody? This might deserve a brief explanation.
Response:
Thank you for this comment. According to your suggestion, we excluded the interpretation at this point and changed the sentence to the following (line 282 – 285):
“However, after co-stimulation of PBMCs with monoclonal anti-CD3 and anti-CD28 antibodies an impaired signal separation of CD3 positive and CD3 negative cells was observed in the presence of MSCs (Figure 2A).”
Same section, paragraph 2: Suggest showing directly that CD5 enhances differentiating CD4+ and CD8+ T cell subsets from other Leukocytes. Concordance between CD3 and CD5 could simply rfelect a lack of differentiating power between the two epitopes.
Response:
You are absolutely right. Stimulation of PBMCs with anti-CD3/CD28 blocks (“masks”) the CD3 antigen, which results in an impaired CD3 signal. It has been discovered that CD5, but not CD3 (Leu-4), is expressed on different B cell lymphomas [1]. As CD3 was insufficient to accurately discriminate between CD3 positive and CD3 negative cells in our set-up, we used CD5 as one of a classical T cell marker, instead. By adding the CD4 and CD8 to the panel, we ensure that no B cells are included in our analysis, as B cells do not express either of those antigens. As shown in Figure 2B, less than 10% of the total PBMC population were CD5 and CD4 or CD8 positive on day 4 and thereafter, which might include B cells, NK cells and CD4/CD8 negative T cells. Our aim in this experiment was not to demonstrate the specificity of either CD3 or CD5 as T cell marker since both of them have already been widely accepted. Instead, we intended to show that gating via CD5 is equivalent to gating via CD3, when CD4 and CD8 are present in the panel. It is highly possible that there is low differentiating power between the two epitopes, as they are co-expressed [2,3].
- Burns, B.F.; Warnke, R.A.; Doggett, R.S.; Rouse, R.V. Expression of a T-Cell Antigen (Leu-1) by B-Cell Lymphomas. Am. J. Pathol. 1983, 113, 165–171.
- Osman, N.; Ley, S.C.; Crumpton, M.J. Evidence for an Association between the T Cell Receptor/CD3 Antigen Complex and the CD5 Antigen in Human T Lymphocytes. Eur. J. Immunol. 1992, 22, 2995–3000, doi:10.1002/eji.1830221135.
- Osman, N.; Lazarovits, A.I.; Crumpton, M.J. Physical Association of CD5 and the T Cell Receptor/CD3 Antigen Complex on the Surface of Human T Lymphocytes. Eur. J. Immunol. 1993, 23, 1173–1176, doi:10.1002/eji.1830230530.
Submission Date
07 February 2023
Date of this review
14 Feb 2023 15:22:17